# Canonical Wnt Pathway Is Involved in Chemoresistance and Cell Cycle Arrest Induction in Colon Cancer Cell Line Spheroids

**DOI:** 10.3390/ijms24065252

**Published:** 2023-03-09

**Authors:** Angela Patricia Moreno-Londoño, María Cristina Castañeda-Patlán, Miguel Angel Sarabia-Sánchez, Marina Macías-Silva, Martha Robles-Flores

**Affiliations:** 1Departamento de Bioquímica, Facultad de Medicina, Universidad Nacional Autónoma de México (UNAM), Mexico City 04510, Mexico; 2Instituto de Fisiología Celular, Universidad Nacional Autónoma de México (UNAM), Mexico City 04510, Mexico

**Keywords:** canonical Wnt signaling, colon cancer, cancer stem cells, chemoresistance, quiescence induction, tumor spheroids

## Abstract

The presence of cancer stem cells (CSCs) has been associated with the induction of drug resistance and disease recurrence after therapy. 5-Fluorouracil (5FU) is widely used as the first-line treatment of colorectal cancer (CRC). However, its effectiveness may be limited by the induction of drug resistance in tumor cells. The Wnt pathway plays a key role in the development and CRC progression, but it is not clearly established how it is involved in CSCs resistance to treatment. This work aimed to investigate the role played by the canonical Wnt/β-catenin pathway in CSCs resistance to 5FU treatment. Using tumor spheroids as a model of CSCs enrichment of CRC cell lines with different Wnt/β-catenin contexts, we found that 5FU induces in all CRC spheroids tested cell death, DNA damage, and quiescence, but in different proportions for each one: RKO spheroids were very sensitive to 5FU, while SW480 were less susceptible, and the SW620 spheroids, the metastatic derivative of SW480 cells, displayed the highest resistance to death, high clonogenic capacity, and the highest ability for regrowth after 5FU treatment. Activating the canonical Wnt pathway with Wnt3a in RKO spheroids decreased the 5FU-induced cell death. But the Wnt/β-catenin pathway inhibition with Adavivint alone or in combination with 5FU in spheroids with aberrant activation of this pathway produced a severe cytostatic effect compromising their clonogenic capacity and diminishing the stem cell markers expression. Remarkably, this combined treatment also induced the survival of a small cell subpopulation that could exit the arrest, recover SOX2 levels, and re-grow after treatment.

## 1. Introduction

Colorectal cancer (CRC) is one of the cancers with the highest incidence (occupying the third position) and the second position in mortality percentage worldwide (Globocan 2020). 5-Fluorouracil (5FU) remains the most common chemotherapeutic drug used in the fight against CRC, either alone or in combination with other medications. Although patients show an initial response to 5FU-based therapy, many of them no longer respond; residual tumoral cells re-gain proliferative capacity and repopulate the tumor, resulting in disease relapse [1,2,3]. In this regard, it has been reported that more than 40% of patients show resistance to 5FU [4], but the mechanisms involved in resistance promotion in residual cancer cells are not fully understood [4,5].

Cancer stem cells (CSCs) represent a cell subset with characteristics similar to healthy stem cells, such as self-renewal capacity, but with tumor-initiating ability and high invasive capacity. One of the most important CSC properties is their ability to resist conventional therapies [6]. Among the mechanisms that allow them to resist treatment are the high expression of multidrug resistance genes, such as the ATP-binding cassette transporter proteins, increased anti-apoptotic protein expression, and more efficient DNA damage repair machinery. But probably the most efficient way to resist therapy mainly directed to eliminate proliferating cells is their ability to enter a quiescent state, in which, apart from this, the tumoral cell can avoid DNA damage induced by the conventional antiproliferative agents [7]. Finding new ways to eradicate the CSCs subpopulation is then of paramount importance. Still, it demands understanding the cellular and molecular mechanisms that allow CSCs to face cytotoxic stress.

The Wnt signaling is one of the main pathways regulating stemness in healthy and cancer cells [8,9]. Wnt ligands in the stem cell niche can activate different Wnt pathways, canonical β-catenin-dependent, and non-canonical β-catenin-independent signaling pathways. Wnt pathways control cell proliferation, survival, migration, and invasiveness. But only the canonical pathway has been directly linked to the maintenance of the stem cell phenotype in CRC. The non-canonical pathway’s role in cancer stem cells remains poorly understood [10]. Experimental evidence in different cancers has shown that canonical Wnt inhibition may sensitize tumoral cells to conventional therapy [11,12], which allows for overcoming resistance. On the other hand, it has also been reported that aberrant Wnt signaling confers resistance to targeted or standard anti-cancer therapies by several mechanisms. These include maintaining the cancer stem cell population, favoring transcriptional plasticity, improving DNA damage repair, and inducing immune evasion [13].

The main objective of this study was to investigate the role played by the canonical Wnt/β-catenin pathway in CSCs resistance to 5FU treatment in colon tumor spheroids, an in vitro model characterized by the enrichment of cells with a stemness phenotype. Our findings indicate that 5FU alone induces in all CRC spheroids tested cell death, DNA damage, and quiescence, but in different proportions for each one. We also found that the activation of canonical Wnt signaling promotes survival in different spheroid cell contexts, which counteracts the negative effects induced by 5FU. Notably, in CSCs with aberrant canonical Wnt activation, the combined treatment of 5FU with a canonical Wnt inhibitor produced severe cytostatic effects via the induction of a G0 arrest and G2 phase accumulation. Despite this, a small CSC population could exit the arrest and re-grow after treatment. Our data suggest, therefore, that canonical Wnt signaling is involved in inducing CSCs resistance to 5FU treatment, but that in addition, other mechanisms besides canonical Wnt signaling are involved in CSCs survival promotion.

## 2. Results

### 2.1. Tumor Spheroid Culture Is Enriched in CSCs and Serves as a Model for CSCs Study

Spheroids culture has been widely used to study cancer stem cells since their enrichment has been demonstrated in these cultures that favor the stem cells’ ability for anchorage-independent growth in a clonal density and serum-free medium [14,15,16]. Many studies have reported that CRC cancer cells in spheroids express markers associated with stemness phenotypes such as CD133, CD44, CD44v6, CD166, and Lgr5 [17,18,19,20], the latter widely used to study CRC cancer stem-like cells both in in vitro and in vivo models [21,22]. To validate our model, we compared the cell surface expression of Lgr5, CD133, CD44, and Cd44v6 between monolayer cultures (2D cultures) and second-generation spheroids on the 11th day of forming spheroids. We found that spheroids showed a heterogeneous expression of these CSCs markers depending on the CRC cell line tested. Overall, spheroids exhibit an increased expression of Lgr5 and CD133 at the cell surface (Figure 1C and Figure 1F, respectively), while the total levels of CD44 and CD44v6 did not change compared with cells grown in 2D cultures (Figure 1B and Figure 1E, respectively). Remarkably, SW480 and RKO spheroids showed a higher proportion of CD44+Lgr5+ double-positive subpopulation than 2D cultures (Figure 1A,D). In SW620 spheroids, we only observed an increase of 2.5-fold in the Lgr5+CD44− subpopulation compared to 2D culture, without showing changes in Lgr5 and CD44 total expression levels. The meaning of these changes in subpopulation distribution in SW620 spheroids compared to 2D cultures is unclear. Several studies have reported that CSCs surface marker expression is dynamic and can fluctuate in response to the tumoral microenvironment [23,24,25,26].

Additionally, we examined the self-renewal ability of the cells cultured in spheroids by assessing spheroid formation efficiency through serial passages (four generations). As shown in Figure 1H, the spheroids cultures from all the cell lines tested contain a subpopulation with unlimited self-renewal ability, which is consistent with the CSC phenotype [16]. The spheroid-forming efficiency (SFE) of RKO and SW480 cell lines increased from the second generation, but only SW480 cells exhibited a significant increase. Although SW620 cells showed a slight reduction in SFE along the passages, it was not statistically significant. These results indicated that spheroids culture is an optimum in vitro model to study cancer stem-like cells.

### 2.2. 5FU Induces Cell Cycle Arrest in CRC Spheroids

To evaluate the 5FU-induced cytotoxicity in CRC spheroids, we treated established spheroids with increasing concentrations of 5FU for 72 h. Figure 2A shows that both SW480 and SW620 spheroids show resistance to 5FU, while RKO spheroids exhibit sensitivity according to cell viability analysis. 5FU induced cell death and DNA damage in the spheroids of all cell lines tested, but RKO spheroids showed higher levels of cleaved-PARP (c-PARP) and p-γH2AX (Ser139) as apoptosis and DNA damage markers, respectively (Figure 2B), followed by SW480 spheroids and SW620 spheroids. Regardless of the mutational status of cell lines, 5FU increased the proportion of cells in a quiescent state (Go phase of the cell cycle) as measured by the non-expression of Ki67, a cell proliferation marker (Figure 2C). Moreover, RKO spheroids showed a decrease in the G2/M phase after 72 h of 5FU treatment (*p* < 0.001) as a consequence of the cell cycle arrest (Figure 2C). In addition, p-CDK2 levels decreased, and p21 levels increased significantly in RKO spheroids due to increasing levels of p53 (wild-type) in response to 5FU-induced DNA damage (Figure 2D). Surprisingly, we detected high basal levels of p21 in SW480 spheroids compared to SW620 spheroids, and 5FU treatment only decreased them in SW480 spheroids. Thus, these data suggest that 5FU-induced cell cycle arrest in Go in SW480 and SW620 spheroids is probably via other CDK inhibitors (Figure 2D). These results could explain a minor capacity for regrowth post-5FU treatment (5FU-free time) under anchorage-dependent conditions (Figure 3C) and the absence of cell division under anchorage-independent conditions after 21 days of incubation (Figure 3B). In addition, as can be seen in Figure 2D, SW620 spheroids were the only ones that showed an increase in p-CDK2 levels. This result was consistent with the higher clonogenic capacity they displayed under anchorage-dependent conditions (Figure 3C) and the emergence of several clones observed on the 21st day of incubation under anchorage-independent conditions during 5FU post-treatment time compared to other cell lines (Figure 3B).

Additionally, because of the detection of low or no regrowth as spheroids in all cell lines after 5FU treatment, we evaluated cell viability on the 21st day of incubation. We found that only 18% of RKO cells and 27% of SW480 cells were still viable, in contrast with 66% of SW620 cells (Figure 3B). This result, along with the limited size of clones of SW480/SW620 and the non-proliferation of RKO cells at the end of the assay compared to established spheroids derived from control cells on the 7th-day post-treatment, reflect a deep cell cycle arrest in RKO cells and a slow-cycling phenotype of survival cells during post-treatment time in SW480 and SW620 cells.

Altogether these results indicated that the metastatic SW620 spheroids display the highest resistance to 5FU, as they exhibit lower cell death and DNA damage compared to spheroids of the other cell lines. In addition, and remarkably, their capacity to re-grow and survive post-treatment is much higher than that of RKO and SW480 tumor spheroids. These abilities would be related to a greater probability of recurrence.

### 2.3. Wnt Pathway Stimulation Promotes Cell Survival in 5FU-Sensitive Spheroids

Given the increasing evidence suggesting the participation of the canonical Wnt pathway in drug chemoresistance [27], we investigated whether Wnt signaling activation by Wnt ligands favors the survival of 5FU-treated spheroids. Because the RKO spheroids do not have a constitutive, ligand-independent canonical Wnt signaling activation and are responsive to Wnt ligands, we treated RKO spheroids with 300 ng/mL of either Wnt3a or Wnt5a ligands to evaluate their effect on 5FU-induced cytotoxicity. We have previously reported that the Wnt3a ligand acts as a prototype canonical ligand, activating the β-catenin transcriptional activity in RKO 2D cell culture [28]. As can be observed in Figure 4A, we also detected that only Wnt3a, but non-Wnt5a, induced the β-catenin transcriptional activity in a subpopulation of spheroids transduced with a GFP-Wnt reporter gene after 24 h of treatment. Moreover, Wnt3a caused an increase in active β-catenin (non-phosphorylated) levels at 24 h of treatment with a subsequent fall at 48 h (Figure 4B). Consistent with this, we detected the increased expression of canonical β-catenin/TCF target genes, such as c-Myc, cyclin D1, and survivin (Figure 4B). Of note, both Wnt3a and Wnt5a increased the level of these proteins, but at different times, at 24 and 48 h, respectively (Figure 4B). Consistent with our previous reports [28], we also observed that while Wnt3a can simultaneously activate canonical and non-canonical Wnt/Ca^2+^ pathways, Wnt5a acts as a non-canonical-prototype ligand in RKO spheroids. Wnt5a could activate Wnt/Ca^2+^ pathway effectors such as the NFAT transcription factor family involved in cell cycle progression, apoptosis, and survival [29,30].

Because we observed that Wnt ligands showed a different response over time according to changes in the protein levels of the target genes mentioned above, we used a different treatment protocol for each Wnt ligand in RKO spheroids. Given that Wnt3a promotes the expression of β-catenin/TCF target genes in the short term (24 h), we treated RKO- spheroids with an acute Wnt3a dose 1 h before the addition of 5FU (co-treatment model). In contrast, Wnt5a treatment was performed 48 h before 5FU treatment (pretreatment model) due to increased levels of proteins evaluated at 48 h. We found that Wnt5a did not reduce the cell death induced by 5FU, measured by c-PARP levels, and Wnt3a only decreased it by 20% (Figure 4C). But importantly, we found that both Wnt ligands increased the expression of anti-apoptotic proteins such as survivin and Bcl-2 (Figure 4D) compared to RKO spheroids treated with 5FU alone. These data, therefore, indicate that canonical Wnt pathway activation, either by ligand (in RKO cells) or already constitutively active (in SW480 and SW620 cells), induces the same effect in colon cancer stem cells: it counteracts the negative effects produced in them by 5FU. This explains why SW480 and SW620 cells are already resistant to 5FU, and RKO cells are sensitive unless Wnt3a stimulates canonical Wnt in them.

### 2.4. Canonical Wnt Inhibition Favors Survival against 5FU-Induced Cytotoxicity in Spheroids with Constitutive Active Canonical Wnt Signaling

The results obtained with RKO spheroids showing that canonical Wnt signaling activation attenuates 5FU-induced cell death suggested that inhibiting canonical Wnt signaling in cells with constitutive activity could sensitize them to chemotherapy. To test this hypothesis, first, we overexpressed a dominant-negative TCF4 (dnTCF4) in SW480 and SW620 cell lines by stable transfection with the pPGS dnTcf-4(deltaN41) plasmid. As can be observed in Figure 5, the inhibition of the β-catenin transcriptional activity mediated by the expression of the dnTCF4 counteracted the negative effect produced by 5FU alone in cell viability (Figure 5B). However, although there was an initial depletion in the β-catenin TOPflash reporter activity in SW480 cells, we observed that it was restored along the passages, possibly due to a compensatory effect by the presence of other TCF/LEF proteins in these cells, such as TCF1 [31]. Nevertheless, this effect was less in the SW620 cell line, and we observed a low β-catenin/TCF transcriptional activity (Figure 5A) sustained along several passages. For that reason, we only evaluated the effect of dnTCF4 expression in SW620 spheroids against 5FU-induced cytotoxicity. We found that dnTCF4 expression decreased the loss of cell viability (Figure 5B) and reduced by 38% the 5FU-induced p-yH2AX levels, with no changes in p-CDK2 and survivin levels (Figure 5C,D) after 72 h of 5FU treatment compared to 5FU-treated control spheroids derived from cells transfected with an empty vector. Analysis of cell cycle distribution did not reveal significant changes in dnTCF4 spheroids treated with 5FU in contrast to control spheroids, apart from a G2/M phase reduction (Figure 5E). These results suggest a slight 5FU effect on the proliferation of SW620 spheroids that express dnTCF4, but we cannot rule out the possibility of a compensation effect from other expressed TCF proteins during spheroid formation.

To investigate the depletion impact of the other TCF proteins, we used the pharmacological inhibitor Adavivint (SM04690), a potent Wnt signaling inhibitor that acts as an ATP competitive small molecule that reduces the expression of TCF1, TCF4 (TCF7L2), AXIN2, and LEF1 at a post-transcriptional level through inhibition of intranuclear kinases CLK2 and DYRK1A, thus leading to an overall inhibitory effect on canonical Wnt-related gene expression [32]. First, we evaluated the efficiency of this inhibitor in SW480 and SW620 cell lines in 2D and spheroids cultures. The results presented in Appendix A show that monolayer cultures of both cell lines treated with increasing Adavivint concentrations (0.03, 0.1, and 1 μM) for 48 h showed a decrease in the β-catenin, TCF1, and TCF4 expression, and also in Wnt target gene expression such as c-Myc and Axin2, with a higher fall in all levels of the proteins evaluated with 1 μM Adavivint. In addition, we observed a reduction in β-catenin, TCF1, and c-Myc expression with 1 μM Adavivint at 72 h of treatment in SW620 spheroids and SW480 spheroids without changes in TCF4 levels (Appendix A). The β-catenin/TCF transcriptional activity also was assessed in spheroids of SW480^TOP-GFP^ cells, obtaining that 1 μM Adavivint decreased 80% of the GFP fluorescence after 72 h of treatment (Appendix A).

According to these results, we pretreated SW480 spheroids and SW620 spheroids with 1 μM Adavivint for 72 h, followed by 5FU treatment for the other 72 h. Surprisingly, the Adavivint pretreatment reduced the apoptosis induced by 5FU (visualized as cleaved-PARP) in spheroids of both cell lines (Figure 6A). Adavivint decreased the DNA damage measured as p-γH2AX (Ser139) levels in SW620 spheroids with no effect on SW480 spheroids (Figure 6B). As a result of canonical Wnt signaling inhibition, the expression of survivin, an anti-apoptotic protein, was downregulated regardless of the 5FU effect (Figure 6C). Therefore, the data indicate that canonical Wnt signaling inhibition by Adavivint enhances survival in SW480 and SW620 spheroids against 5FU-induced damage. In addition, Adavivint pretreatment decreased the p-CDK2 levels in spheroids of both cell lines but only caused a significant increase in p21 in SW620 spheroids (3.5-fold compared to control), whereas 5FU-induced p21 depletion in SW480 spheroids was not significantly decreased by Adavivint (Figure 6D). These results indicate that canonical Wnt inhibition could disturb the cell cycle and impose a slow-proliferative phenotype.

We next analyzed the cell cycle distribution only in cells from SW620 spheroids due to the overall impact of Adavivint on 5FU-chemoresistance, the metastatic cell line that exhibits higher resistance to 5FU compared to spheroids of the other cell lines, as we described before. Adavivint pretreatment did not induce changes in quiescent cell subpopulation (Go phase) compared to spheroids treated with 5FU alone, as we expected. In contrast, independent of the 5FU stimulus, Adavivint treatment increased the proportion of cells in the S and G2/M phases (Figure 6E). Considering the reduction in p-CDK2, a proliferation marker, and the increase in p21 in SW620 spheroids in conjunction with the changes in the cell cycle distribution, the data suggest that Adavivint causes a delay in the cell cycle at the S and G2 phases compared to the control, rather than reinforce the cell cycle arrest induced by 5FU. G2-phase delaying would favor DNA repair before mitosis and improve survival under endogenous and 5FU-induced damage [33,34,35].

### 2.5. Canonical Wnt Signaling Inhibition Induces a Reduction in CSCs Markers Expression and Clonogenic Capacity in Tumor Spheroids

To evaluate whether canonical Wnt pathway inhibition affects the proportion of the CSC subpopulation, we determined the expression of cancer stem cell markers such as CD133 and Lgr5 that increased in spheroid cultures, as described before (Figure 1). We found that the inhibitors Adavivint and 5FU (alone or in combination) induced depletion in CD133 expression without affecting the Lgr5 and CD44v6 levels in SW620 and SW480 spheroids (Figure 7A,B and Appendix A). Moreover, we measured the levels of Oct4 and Sox2, transcription factors involved in the maintenance of embryonic stem cell phenotype [36]. We observed that Adavivint only reduced the Oct4 levels in SW480 spheroids but decreased the levels of Sox2 in both cell lines (Figure 7C,D and Appendix A) independent of 5FU treatment and particularly when we used the combined Adavivint + 5FU treatment. Consistent with the reduced expression of some cancer stem markers, we found that combined treatment induced the expression of cytokeratin 20 (CK20) in SW620 spheroids (Figure 7E), a differentiation marker of the intestinal epithelium. However, we detected the opposite effect in SW480 spheroids. Thus, data indicate that Adavivint induced a different degree of response in spheroids of SW480 and SW620, probably due to alternative mechanisms of activation.

Because Adavivint reduced the expression of cancer stem cell markers, we used a clonogenic assay to assess its impact on stemness-relevant phenotypes. We treated spheroids with Adavivint for six days, seeded cells from dissociated spheroids under anchorage-dependent conditions, and fixed colonies after eight days of growth. Adavivint reduced the colony formation efficiency of SW620-derived cells from spheroids and depleted the clonogenic capacity of SW480 cells in contrast to cells derived from control spheroids (Appendix A). Considering that Adavivint had a pronounced effect on SW480 clonogenicity, we only evaluated Adavivint + 5FU’s effect on SW620 clonogenicity. We observed a similar trend when we used Adavivint combined with 5FU, but the clonogenicity reduction did not reach significance compared to 5FU alone (*p* = 0.0589) (Figure 7F). In addition, we seeded cells in low-attachment microplates to evaluate their spheroid-forming capacity after treatment with 5FU alone or in combination with Adavivint. We followed the cultures for up to 40 days when we observed established spheroids. Although we found a depletion in the number of spheroids with the use of both drugs compared to 5FU treatment alone (Figure 7G), we detected an increase in Sox2 levels using the combined treatment, which suggests an increase in proliferation rates, consistent with the emergent survival subpopulation with the ability to form spheroids (Figure 7H).

In brief, our data indicate that canonical Wnt signaling activation is involved in sustaining CSCs resistance to 5FU treatment. However, although the inhibition of the canonical β-catenin signaling also improved the survival of spheroids treated with 5FU, it adversely impacted the stemness phenotype, apparently in dependence on the malignancy grade of cancer cells. Remarkably, we found here that combined treatment with 5FU and Adavivint also induced the survival of a small cell subpopulation, probably through a combined effect on the cell cycle, promoting G0 arrest and G2 phase accumulation. This subpopulation could exit the arrest and re-grow after treatment, suggesting, therefore, that other mechanisms participate in resistance induction to 5FU treatment of CSCs.

## 3. Discussion

Aberrant Wnt signaling is a hallmark of most colorectal cancers. Therefore, targeting the Wnt-signaling pathway has been the focus of cancer research for a long time. The presence of cancer stem cell subpopulations in tumors is the hub of resistance generation. Yet, little is known about the mechanisms that allow CSCs to switch to a latent state in response to stressful environmental conditions, to remain quiescent while retaining their tumor-initiating capacity, and to exit from this state to evolve into aggressive metastasis. Although several Wnt pathway inhibitors have been described with promising efficacy in therapy, how the Wnt pathway is involved in CSCs chemoresistance is not clearly established.

In this work, we investigated the role of the canonical Wnt/β-catenin pathway in CSCs chemoresistance. We demonstrate that 5FU induces cell death and DNA damage in different grades in the spheroids of the CRC cell lines used. Additionally, 5FU induced the entry to quiescence in a cell subset in spheroids of all cell lines tested, independent of their Wnt signaling status. RKO spheroids exhibited the most responsiveness to 5FU, whereas the metastatic SW620 spheroids exhibited 5FU-chemoresistance, as determined by less cell death and DNA damage at the end of treatment compared with SW480 and RKO spheroids.

Quiescence (G0 phase) is considered a chemoresistance mechanism by which the tumoral cell can evade the cytotoxic action of drugs that target highly proliferating cells [37]. Our data indicate that 5FU treatment induces DNA damage and cell death and, importantly, increases the proportion of CSCs in a quiescent state. However, the capacity of cells to re-grow depends on their ability to exit from quiescence in post-treatment time, resulting in a transient or deep quiescent state. Intrinsic and extrinsic signals are determinants for switching into a proliferating state, such as mitogen levels and the balance between CDK-cyclin complexes and CDK inhibitors (p21, p27, and p57) at the end of the mitotic phase [38,39].

To investigate the role of the canonical Wnt pathway in 5FU-chemoresistance, we used two methodological approaches: (1) Wnt ligand stimulation in RKO spheroids that exhibit normal canonical Wnt signaling to determine whether Wnt signaling activation promotes survival during 5FU treatment. (2) Constitutive active Wnt signaling inhibition in spheroids of cell lines that exhibit aberrant canonical Wnt activation (SW480/SW620). In the first approach, our data clearly showed that canonical Wnt pathway activation, whether induced by ligand (in RKO cells) or already constitutively active (in SW480 and SW620 cells), produced the same effect in colon cancer stem cells: it counteracted the negative effects produced in them by 5-FU. This explains why SW480 and SW620 cells are already resistant to 5-FU and RKO cells are sensitive unless Wnt3a stimulates canonical Wnt in them.

Consistent with our results, several studies have shown that Wnt ligand-dependent β-catenin/TCF activation contributes to chemoresistance through induction of its target genes such as survivin, ABC transporters, proteins involved in DNA repair, and proteins involved in cell cycle progression [27,40], which allow cells to survive against cytotoxic stress and re-grow once the intrinsic and extrinsic conditions are optimal. In this regard, Cho et al. reported that 5FU promotes stemness in residual tumoral cells through Wnt3a/β-catenin pathway activation via p53 transcriptional induction in CRC cells harboring wild type p53 [41], resulting in relapse after chemotherapy. Combinatorial treatment with Wnt inhibitor (Porcupine inhibitor) and 5FU abrogated CSCs activation and reduced tumor re-grow during post-treatment (drug free). This evidence, therefore, would be consistent with the hypothesis that canonical Wnt signaling inhibition is a potential therapeutic strategy to inhibit the acquisition of cancer stem phenotype and overcome recurrence after standard chemotherapy. However, contrary to this hypothesis, we found here that the inhibition of the β-catenin transcriptional activity, either mediated by the expression of a dnTCF4, or using the inhibitor Adavivint in combination with 5FU also counteracted the negative effect produced by 5-FU in cell viability, and enhanced survival in SW480 and SW620 spheroids against 5FU-induced damage. However, although the inhibition of the canonical β-catenin signaling improved the survival of spheroids treated with 5FU by decreasing DNA damage and proliferation, it adversely impacted the stemness phenotype through impaired Sox2 and even Oct4 expression, apparently in dependence on the malignancy grade of cancer cells. Remarkably, we found that combined treatment with 5-FU and Adavivint also induced the survival of a small cell subpopulation, probably through a combined effect on the cell cycle, promoting G0 arrest and G2 phase accumulation. This subpopulation persisted after 40 days of culture and remarkably showed an increase in the expression levels of the stem cell marker Sox2, which suggests an increase in proliferation rate, consistent with the emergence of this survival subpopulation with the ability to form spheroids. In agreement with our results, very recently Alvarez-Varela A. and Cañellas-Socias A. from the Eduard Batlle group [42,43] have revealed that drug-tolerant persister colorectal cancer cells that mediate relapse after chemotherapy downregulate the Wnt/stem cell gene program immediately after chemotherapy. Interestingly, while widespread evidence has demonstrated that CRC growth is driven by a subset of LGR5+ stem-cell-like tumor cells, their analyses revealed that high-resistant cancer cells (HRCs) represent a distinct cell population. Their findings revealed that the adaptation of cancer stem cells to suboptimal niche environments protects them from chemotherapy and identifies a candidate cell of origin of relapse after treatment in CRC that they called HRC. Similar findings have been obtained by Wu C. from the Sara M. Weis and David A. Cheresh group, studying how microenvironmental stresses affect tumor initiation and progression of aggressive pancreatic cancer cells [44]. These authors have recently reported that the harsh microenvironments of the body can push certain pancreatic cancer cells to overcome the stress of being isolated and make them more adept at initiating and forming new tumor colonies. And interestingly, they also observed that while tumor-initiating cells or cancer stem cells (TICs) may inherently possess resistance to such stresses, they showed that a few non-TICs might undergo adaptive reprogramming, giving rise to new colonies that will eventually outgrow them.

Regarding the differences observed in some results obtained with the Adavivint compared with the expression of the dnTCF4, we observed that the Adavivint produced a more severe effect on the expression of cell proliferation markers such as p-CDK2 compared to the effect obtained with only the dnTCF4, whose expression did not affect the increase induced by 5FU or the survivin levels. It must be taken into account that Adavivint is a potent inhibitor of canonical Wnt-related gene expression via a β-catenin-independent mechanism, mainly through CLK2 and CLK3 inhibition resulting in unstable transcripts of Wnt-related genes. It reduces not only the expression of Wnt target genes such as c-Myc and survivin but also the expression of Frizzled Wnt receptors, Dvl proteins, and TCF/LEF transcription factors [32,45]. Thus, Adavivint may alter not only the TCF4-mediated gene regulation but also the gene regulation mediated by TCF1 and LEF1, avoiding the compensatory effect that these factors could produce when only the TCF4-mediated transcription is blocked by expressing a dnTCF4. However, because Adavivint can inhibit several other kinases and CLK is actively involved in pre-mRNA processing, we cannot rule out the impact on other signaling pathways that could explain the overall anti-tumor mechanism of this drug.

In summary, our data indicate that canonical Wnt signaling activation induces CSCs resistance to 5FU treatment, promoting survival in different spheroid cell contexts, which counteracts the negative effects induced by 5FU. However, although the inhibition of the canonical β-catenin signaling also improved the survival of spheroids treated with 5FU, it adversely impacted their stemness phenotype through impaired Sox2 expression and even Oct4 expression, apparently in dependence on the malignancy grade of cancer cells. Remarkably, we also found that combined treatment with 5FU and Adavivint induced cell cycle arrest. Still, a small cell subpopulation persisted, recovered Sox2 expression levels, and began to re-grow after treatment. The mechanisms involved in this CSCs capacity to exit cell cycle arrest and re-grow remain to be elucidated.

## 4. Materials and Methods

### 4.1. Reagents and Antibodies

The antibodies used in this study include the following: rabbit anti-phospho-H2AX, rabbit anti-cleaved-PARP, rabbit anti-phospho-CDK2, rabbit anti-p21, rabbit anti-active-β-catenin, rabbit anti-survivin, rabbit anti-TCF1, and rabbit anti-Axin2 were all obtained from Cell Signaling Technology (Danvers, MA, USA). Mouse anti-c-Myc, mouse anti-cyclinD1, mouse anti-Bcl2, mouse anti-β-catenin, rabbit anti-TCF4, mouse anti-Sox2, mouse anti-Oct4, and mouse anti-GAPDH were obtained from Santa Cruz Biotechnology (Dallas, TX, USA). Allophycocyanin (APC)-conjugated mouse anti-CD44 were obtained from BD Bioscience. Phycoerythrin (PE)-conjugated mouse anti-CD133, mouse anti-CD44v6, mouse anti-Lgr5, rabbit anti-CD133, and mouse anti-Ki67 were obtained from ThermoFisher Scientific (Waltham, MA, USA). APC-Vio770-conjugated mouse anti-Lgr5 was obtained from Miltenyi Biotec (Bergisch Gladbach, Germany). FITC-conjugated goat anti-mouse was obtained from Jackson ImmunoResearch (West Baltimore Pike, PA, USA). Mouse anti-αTubulin was obtained from Sigma (St. Louis, MO, USA). Goat anti-mouse and anti-rabbit IgG-horseradish peroxidase-conjugated were from Pierce (Rockford, IL, USA). AquaZombie dye was obtained from Biolegend (San Diego, CA, USA). 5-fluorouracil (5FU) was purchased from Sigma-Aldrich (St. Louis, MO, USA). Recombinant human Wnt3a and recombinant human Wnt5a ligands were purchased from R&D Systems (Minneapolis, MN, USA). Adavivint (SM04690) was purchased from Selleckchem (Houston, TX, USA).

### 4.2. Plasmids

The M50 Super 8x TOPFlash and M51 Super 8x FOPFlash reporter plasmids (Addgene 12456 and 12457) were Randall Moon’s gifts. The TOP-GFP-mCherry plasmid used to measure the TOP reporter in spheres was a gift from Ramesh Shivdasani (Addgene 35491). To obtain TOP-GFP reporter cells, lentiviral-transduced cells were cell-single sorted by mCherry fluorescence using the FACsAria sorter and examined for their reporter responsiveness to Wnt ligands (RKO^TOP-GFP^ reporter cells) or their constitutive activity (SW480^TOP-GFP^ reporter cells). The plasmid encoding human dominant-negative TCF4 (dnTCF4) [pPGS dnTcf-4(deltaN41)] was a gift from Eric Fearon (Addgene plasmid 19284).

### 4.3. Cell Lines

All cells were obtained from the American Type Culture Collection (ATCC) (Manassas, VA, USA). They were authenticated in June 2018 by Short Tandem Repeat DNA profiling analysis performed at the Instituto Nacional de Medicina Genómica (INMEGEN) in Mexico City. We used the human RKO and SW480 cell lines isolated from primary cancer, whereas the SW620 cell line was derived from a lymph node metastasis from the same patient that the SW480 cell line was derived from. The RKO cell line is a prototype of BRAF-driven colon cancer cells that express wild-type p53 and exhibit highly inducible canonical Wnt signaling with no basal activation. SW480 and SW620 cell lines are the prototypes of KRAS-driven cancer cells. They express KRAS and p53 mutated proteins (TP53 R273H, KRAS G12V) and display constitutive active canonical Wnt signaling because they express a truncated APC protein [46].

### 4.4. Cell Culture

*Monolayer cultures (2D cultures):* The RKO cell line was maintained in Dulbecco’s Modified Eagle’s Medium (DMEM) supplemented with 10% fetal bovine serum (FBS) and 2mM L-glutamine, with antibiotics (100 U/mL penicillin, 100 μg/mL streptomycin, and 25 μg/mL amphotericin). SW480 and SW620 cells were maintained in DMEM F-12 supplemented with 5% FBS and 2 mM L-glutamine with antibiotics and amphotericin. All cell lines were cultured at 37 °C in an atmosphere of 95% humidity and 5% CO_2_.

*Spheroid cultures (3D cultures):* cells from monolayer cultures were dissociated with trypsin, counted, and seeded at 1 × 10^3^ cells/mL in serum-free medium containing DMEM F-12 supplemented with 1X B27 and 20 ng/mL EGF with antibiotics (100 U/mL penicillin, 100 μg/mL streptomycin and 10 μg/mL gentamicin, and 25 μg/mL amphotericin) in low-attachment plates to form first-generation spheroids. Fresh medium was added each fourth day. To sequential passages, established spheroids, from the 11th day of culture, were collected and dissociated to single cells using 1X TrypLETM Express (Gibco, Waltham, MA, USA) according to manufacturer instructions. Experimental treatments were performed on second-generation spheroids (2G). Spheroid-forming efficiency was measured by counting the number of colonies on the 7th day of the spheroid-forming culture.

### 4.5. Cell Viability

To evaluate the effect of 5FU on the viability of CRC tumor spheres, cells from first-generation CRC spheres were seeded at 1 × 10^3^ cells in 96-well low-attachment plates. On the 11th day of forming spheres, they were treated with different concentrations of 5FU (vehicle, 10, 100, 500, 1000, 2500, and 5000 μM) for up to 72 h. Cell viability was assayed using the CellTiter-Glo 3D Cell Viability Assay Kit (Promega, Madison, WI, USA) according to the manufacturer’s instructions, and luminescence was measured using a multimode microplate reader. 

### 4.6. Western Blotting

Cells were lysed with RIPA buffer (50 mM Tris-HCl, pH 7.4, 150 mM NaCl, 1 mM EDTA, 0.5% sodium deoxycholate, 1% NP-40, 0.1% SDS, and protease and phosphatase inhibitors) for at least 1 h at 4 °C. After centrifugation (15,000× *g* for 10 min), lysates were collected and stored at 70 °C until analysis. Proteins were separated by 8.5, 10, or 13.5% SDS-polyacrylamide gel electrophoresis (SDS-PAGE) and transferred to nitrocellulose membranes (Bio-Rad, Hercules, CA, USA). The membranes were blocked with 50 mg/mL nonfat dry milk or 3% BSA in Tris-buffered saline for 1 h and incubated overnight at 4 °C with the appropriate primary antibodies. The membranes were incubated with the corresponding horseradish peroxidase (HRP)-conjugated secondary antibody for 2 h at room temperature. Detection was achieved using the SuperSignal Kit (Pierce, Rockford, IL, USA) in the C-DiGit Blot scanner (LI-COR Biosciences, Lincoln, NE, USA) and analyzed by Image Studio™ Lite v5.2 Software (LI-COR Biosciences). GAPDH, α-tubulin, or Ponceau Red stain for total protein were used as loading controls.

### 4.7. Flow Cytometry and Cell Cycle Analysis

For cell surface protein staining, dissociated cells were collected by centrifugation at 5000× *g* and incubated with the Aqua Zombie dye (non-permeant to live cells) for 30 min at room temperature in darkness to exclude dead cells during data analysis. Then, the cell pellet was washed with PBS and incubated with anti-CD44-APC-coupled (BD Biosciences, San Diego, CA, USA), anti-CD133-PE-coupled (eBioscience, San Diego, CA, USA), or anti-Lgr5-APC-Vio770-coupled (Miltenyi, Bergisch Gladbach, Germany) antibodies for 30 min at 4 °C in darkness. CD44v6-surface detection was performed by incubating with anti-CD44v6 primary antibody (eBioscience) for 30 min at 4 °C. The cell pellet was washed with PBS, followed by incubation with FITC-coupled anti-mouse secondary antibody for 30 min at 4 °C in darkness. Samples were acquired on an Attune NXT cytometer.

For cell cycle analysis, resting/quiescent cell populations (Go phase) discrimination and quantification of cell cycle distribution were performed by simultaneous analysis of Ki67 (proliferation marker) and cellular DNA content. Dissociated cells were permeabilized with 70% ice-cold ethanol and stained with anti-Ki67 for 30 min at room temperature. After incubation with FITC-coupled anti-mouse secondary antibody, cells were washed with PBS and incubated with RNAse (100 μg/mL) and propidium iodide (20 μg/mL) for 20 min at room temperature in darkness. Cells stained with the secondary antibody alone were used as a negative control. Samples were analyzed on an Attune NXT cytometer. All data obtained were analyzed by FlowJo v10 software (BD, Ashaland, OR, USA).

### 4.8. Anchorage-Dependent and-Independent Growth

Cells from first-generation spheroids were seeded to 1 × 10^3^ cells/mL in low-attachment plates to form second-generation spheroids. On the 11th day of sphere forming, spheroids were treated with a single dose of 5FU or vehicle for 72 h. On the 14th day, the spheroids were dissociated into single cells and seeded at 5000 cells/mL in low-attachment plates and in TC-treated microplates to determine the independent and anchorage-dependent re-grow abilities post-5FU treatment, respectively.

Cells in low-adherence cultures were maintained in a sphere-growth medium (serum-free) for up to 21 days. On the 21st day, cells were collected, dissociated, and viable cell proportion was measured using Aqua zombie dye (non-permeant to live cells). Furthermore, cells that grew under anchorage-dependent conditions were fixed in cold-methanol and stained with crystal violet at the 7, 14, and 21 days of incubation to follow the ability to grow post-5FU treatment. The colony formation efficiency (% CFE) was determined by counting colonies using ImageJ software (version 1.53a obtained from the National Institutes of Health website, http://imagej.nih.gov/ij/).

### 4.9. TOP Reporter Gene Assay

Cells were seeded in 96-well plates at a density of 2.5 × 10^4^ cells per well. At 24 h after seeding, cells were placed in a serum-free medium and transfected with 200 ng of the reporter plasmid (M50 Super 8x TOPFlash) or control plasmid (Super 8x FOPflash) with 20 ng of the pRL luciferase plasmid (transfection control). The reporter activity in the cell lysates was measured 30 h after transfection using the Dual-Luciferase Assay Kit (Promega, Madison, WI, USA), and the activity was normalized with respect to Renilla luciferase activity.

To assess the β-catenin transcriptional activity in spheroids, transduced cells with TOP-GFP-mCherry plasmid previously sorted were seeded to form spheroids in low-attachment plates. Spheroids stimulated or unstimulated were dissociated, and the reporter activity was analyzed by flow cytometry.

### 4.10. Wnt Pathway Inhibition

SW480 and SW620 cell lines that exhibit constitutively active Wnt signaling were treated with different concentrations of Adavivint (0.03, 0.1, and 1 μM) for 48 h in monolayer cultures or 72 h in spheroids on the 9th day of culture.

Additionally, the inhibition of the canonical Wnt pathway was performed by the expression of dnTCF4 in the SW620 cell line. Cells were transfected with pPGS/ dnTcf-4(deltaN41) plasmid, and after 48 h post-transfection, cells were selected with the G418 antibiotic for 15 days. Subsequently, a TOP reporter assay was performed in clones to survive the selection process. Cells transfected with an empty pCDNA3 vector coding for the resistance gene to the G418 antibiotic were used as control cells.

### 4.11. Statistical Analysis

All data is represented as the mean ± standard error of the mean (SEM) of at least three independent experiments. Comparative analysis between the two groups was performed by Student’s *t*-test. Multiple comparisons between three or more groups were performed using a one-way analysis of variance (ANOVA) followed by Bonferroni’s multiple comparison test. A *p*-value of *p* < 0.05 was considered statistically significant.

## Figures and Tables

**Figure 1 ijms-24-05252-f001:**
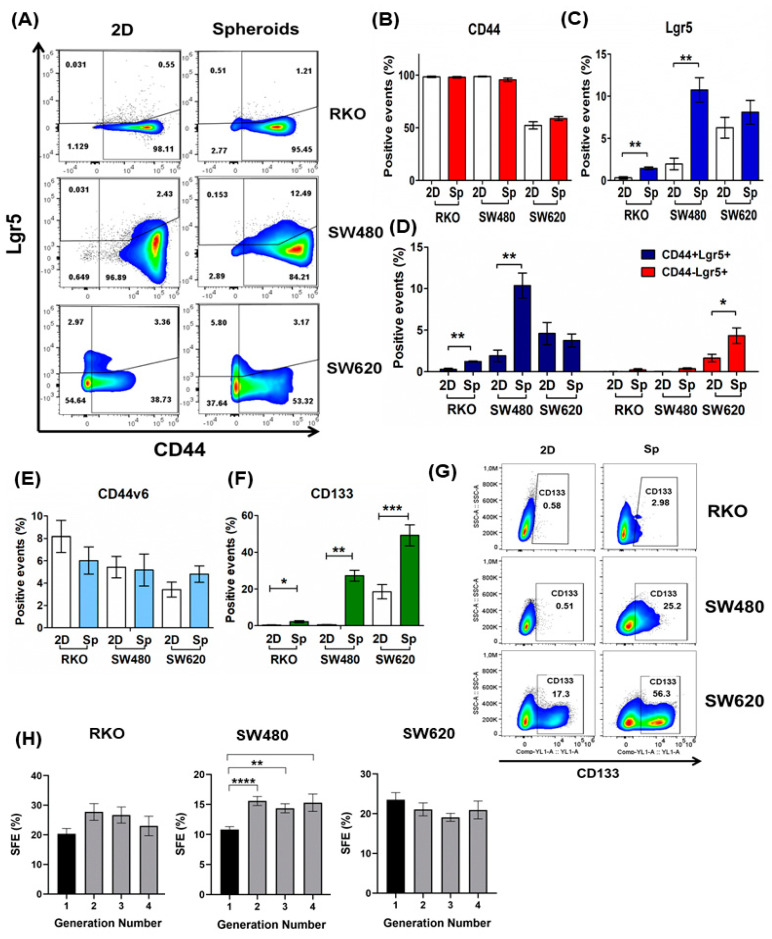
Spheroids as a model of cancer stem cells (CSCs) enrichment of CRC cell lines. (**A**–**G**) Comparative expression of surface markers associated with stemness in second-generation spheroids compared to monolayer cultures (2D) in RKO, SW480, and SW620 cell lines. The proportions of CD44 (**B**), Lgr5 (**C**), CD44v6 (**E**), CD133 (**F**,**G**), CD44+Lgr5+, and CD44−Lgr5+ subpopulations (**A**,**D**) were measured by flow cytometry. (**H**) Quantification of the spheroid formation efficiency (SFE %) of the RKO, SW480, and SW620 cell lines throughout four consecutive generations. Data are represented as the mean values ± SEM of at least three independent experiments. (**B**–**F**) Student’s *t*-test; (**G**) One-way ANOVA followed by Bonferroni’s multiple comparisons. * *p* < 0.05; ** *p* < 0.001; *** *p* < 0.0001; **** *p* < 0.00001.

**Figure 2 ijms-24-05252-f002:**
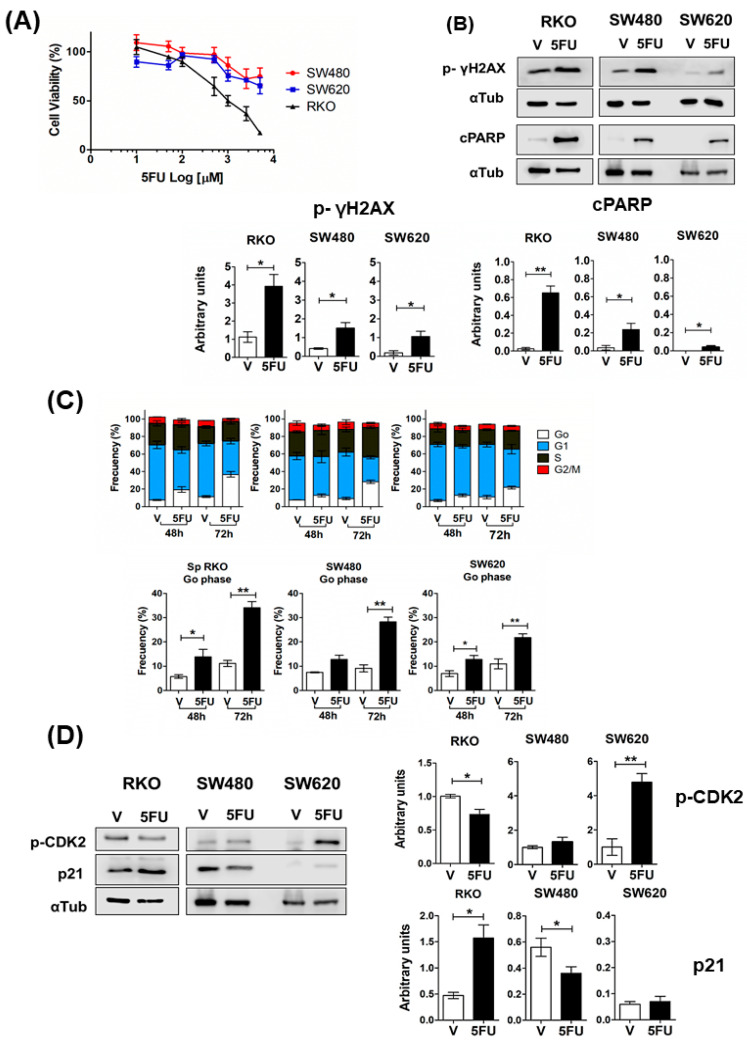
5FU induces cell death and cell cycle arrest in CRC spheroids. (**A**,**B**) 5FU cytotoxicity on spheroids from colon cancer cell lines (RKO, SW480, and SW620). (**A**) Dose–response assay of spheroids treated with 5FU for 72 h. Cell viability was determined using the CellTiter-Glo 3D Kit. (**B**) **Upper** panel: Representative immunoblots of cleaved PARP (cPARP) and phospho-histone γH2AX (Ser 139) in spheroids after 72 h of treatment with 5FU or vehicle (V). α-tubulin was used as a loading control. **Lower** panel: quantification of the relative expression levels of the indicated proteins in the **upper** panel. (**C**) **Upper** panel: cell cycle distribution determined by Ki67 detection and propidium iodide staining in spheroids treated with 5FU for 48 and 72 h. **Lower** panel: a comparative analysis of the Go phase between spheroids treated with or without 5FU at the indicated time points. (**D**) **Left** panel: representative immunoblots of phospho-CDK2 (Thr160) and p21 in spheroids after 72 h of treatment with 5FU or vehicle (V). α-tubulin (αTub) was used as a loading control. **Right** panel: quantification of the relative expression levels of the indicated proteins in the **left** panel. Data are represented as the mean values ± SEM of at least three independent experiments. Student’s *t*-test * *p* < 0.05; ** *p* < 0.001.

**Figure 3 ijms-24-05252-f003:**
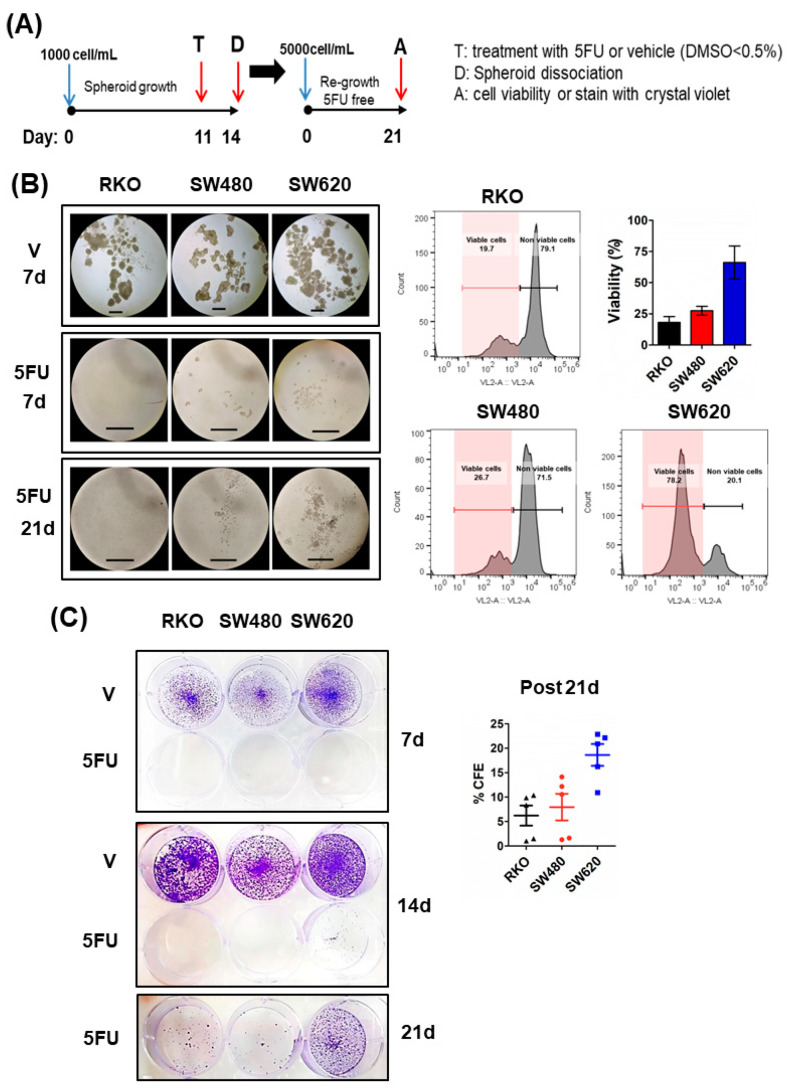
SW620 metastatic cells exhibit an increased capacity for re-grow after 5FU treatment under independent and anchorage-dependent conditions compared to SW480 and RKO cell lines derived from primary tumors. (**A**) Experimental scheme to evaluate the ability of CRC spheroid cells to re-grow after 5FU treatment in anchorage-independent and-dependent conditions compared to cells from spheroids treated with vehicle (V) according to methodology. Spheroids treated were dissociated and seeded to 5000 cells/mL in low-attachment and Tissue culture-treated microplates, respectively. (**B**) **Left** panel: Photographs of cell cultures in low-attachment microplates on the 7th, 14th, and 21st day of incubation after spheroids dissociation and cell re-seeding (scale bar: 500 μm). **Right** panel: Cell viability of collected cells in these cultures was determined using AquaZombie dye on the 21st day of incubation. Fluorescence was measured by flow cytometry. (**C**) Colonies formed in high adherence surfaces (tissue culture-treated microplates) were stained with crystal violet on the 7th, 14th, and 21st days of growth after cell re-seeding. The colony formation efficiency (% CFE) post-5FU treatment was calculated on the 21st day. Images are representative of at least three independent experiments.

**Figure 4 ijms-24-05252-f004:**
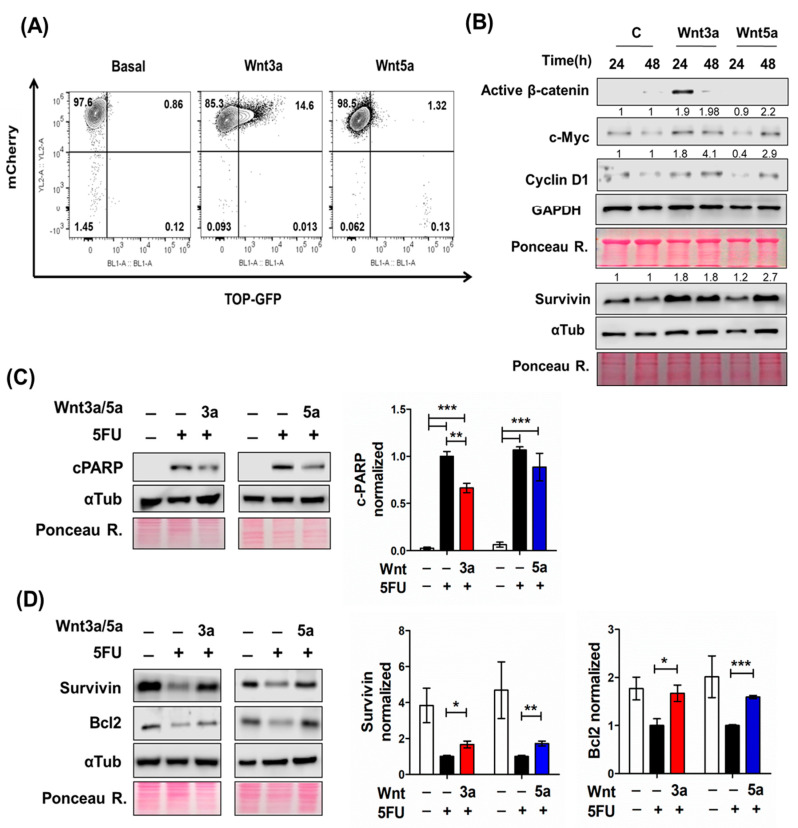
Wnt ligands decrease 5FU-induced death in RKO spheroids through the induction of antiapoptotic protein expression. (**A**,**B**) Wnt3a but non-Wnt5a activate the canonical Wnt/β-catenin pathway in RKO spheroids. (**A**) β-catenin/TCF-mediated transcriptional activity was determined in RKO spheroids from sorted cells that were transduced with double-color construct TOP-GFP.mCherry reporter. Spheroids were treated with 300 ng/mL of Wnt3a or Wnt5a for 24 h, and fluorescence was measured by flow cytometry in dissociated spheroid cells. (**B**) Changes in active (non-phosphorylated) β-catenin, c-Myc, cyclin D1, and survivin levels in RKO spheroids treated with 300 ng/mL of Wnt3a or Wnt5a for 24 and 48 h were determined by Western blot. Ponceau Red staining for total protein was used as loading control. Quantification of the relative levels of indicated proteins was normalized to control. Fold change with respect to control (C 24 h or C 48 h) is shown on the top of each band. (**C**,**D**) **Left** panels: representative immunoblots of the indicated proteins in RKO spheroids treated with or without Wnt ligands under a scheme of pre-treatment or co-treatment with 5FU. Pre-treatment: RKO spheroids were treated with 300 ng/mL of Wnt5a for 48 h and subsequently treated with 5FU for 72 h. Co-treatment: RKO spheroids were treated with 300 ng/mL of Wnt3a ligand 1 h before adding 5FU. Spheroids were collected and lysed after 72 h of 5FU treatment. Ponceau Red staining for total protein was used as a loading control. **Right** panels: quantification of the relative levels of indicated proteins in the **left** panels. Protein levels were normalized to 5FU treatment. Data are represented as the mean values ± SEM of at least three independent experiments. * *p* < 0.05; ** *p* < 0.001; *** *p* < 0.0001.

**Figure 5 ijms-24-05252-f005:**
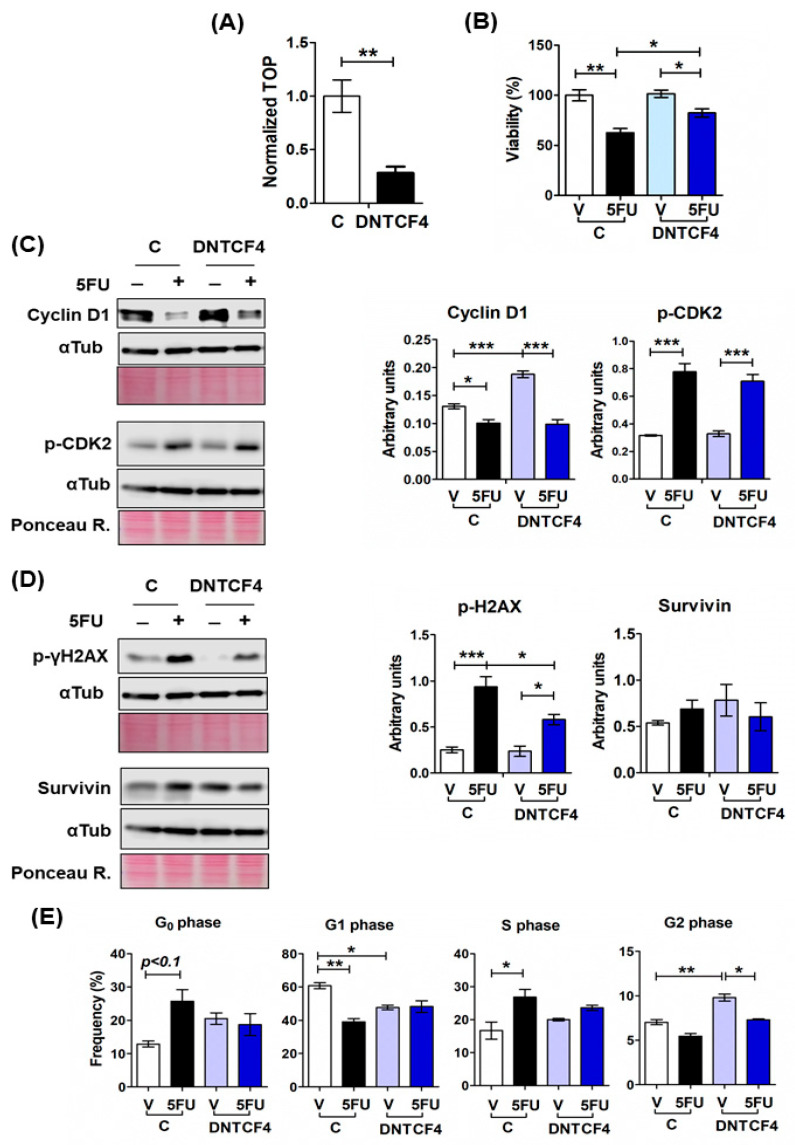
Dominant negative TCF4 (DNTCF4) promotes cell survival in SW620 Spheroids treated with 5FU through decreasing DNA damage. (**A**) β-catenin/TCF-mediated transcriptional activity decreased as a result of DNTCF4 expression in SW620 cells transfected with plasmid pPGS dnTcf-4(deltaN41). The normalized activity of the TCF reporter (TOP-Luc) is shown. Renilla luciferase levels were used as transfection control. (**B**) SW620 spheroids that express dnTCF4 were treated with 5FU for 72 h. Cell viability was measured using the CellTiter Glo-3D Kit and compared to spheroids derived from cells transfected with an empty vector. (**C**,**D**) Immunoblot analysis of cyclin D1 (**C**), phospho-CDK2 (Th160) (**C**), phosphor-histone γH2AX (Ser 139) as DNA damage marker, and survivin (**D**) in SW620 spheroids^dnTCF4^ and control spheroids treated with or without 5FU for 72 h. Ponceau Red staining for total protein was used as loading control. (**E**) Comparative cell cycle analysis of SW620 spheroids^dnTCF4^ and control spheroids treated with or without 5FU for 72 h. Data are represented by the mean ± SEM of at least two independent tests. One-way ANOVA followed by Bonferroni’s multiple comparisons. * *p* < 0.05; ** *p* < 0.001; *** *p* < 0.0001.

**Figure 6 ijms-24-05252-f006:**
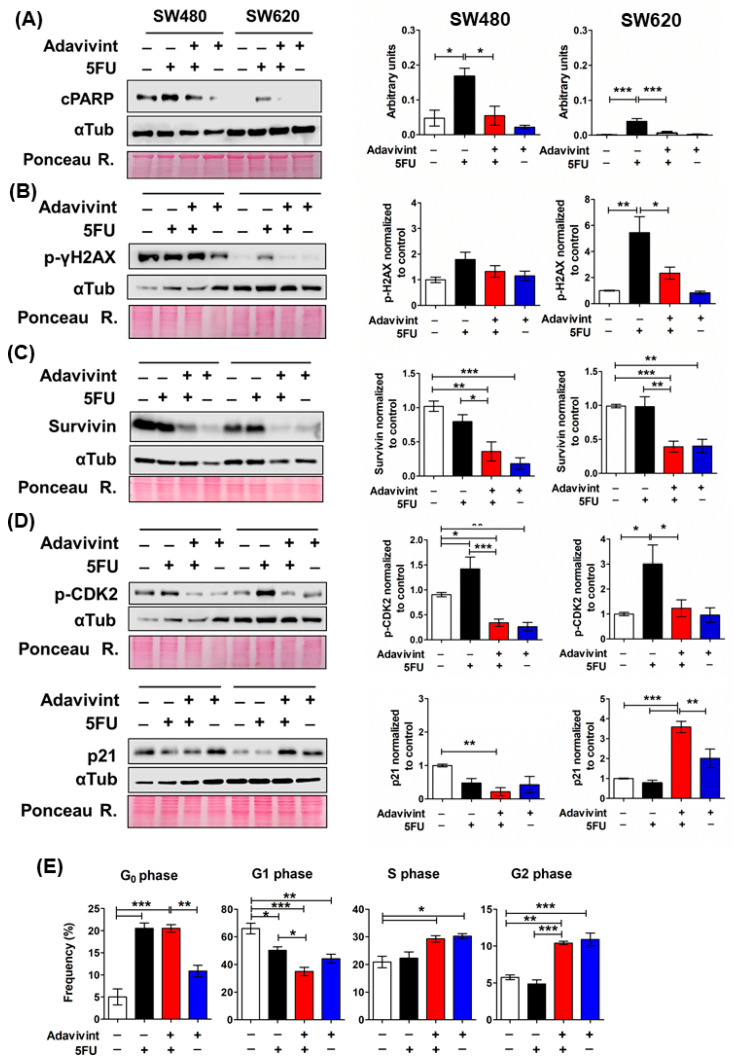
Canonical Wnt pathway inhibition in CRC spheroids increases 5FU resistance by decreasing cell death and inducing cell cycle arrest. SW480 spheroids and SW620 spheroids were pre-treated with or without 1 μM Adavivint for 72 h and subsequently with or without 5FU for 72 h. (**A**–**D**) Representative immunoblots of indicated proteins in SW480 spheroids and SW620 spheroids treated with the same conditions as described in (**A**). Ponceau red staining was used as a loading control (**left** panels). Quantification of the relative levels of indicated proteins was normalized to control-treated with vehicles of Adavivint and 5FU (**right** panels). (**E**) Cell cycle analysis of SW620 spheroids treated with the same conditions as described in (**A**). Data are represented by the mean ± SEM of at least three independent tests. One-way ANOVA followed by Bonferroni’s multiple comparisons; * *p* < 0.05; ** *p* < 0.001; *** *p* < 0.0001.

**Figure 7 ijms-24-05252-f007:**
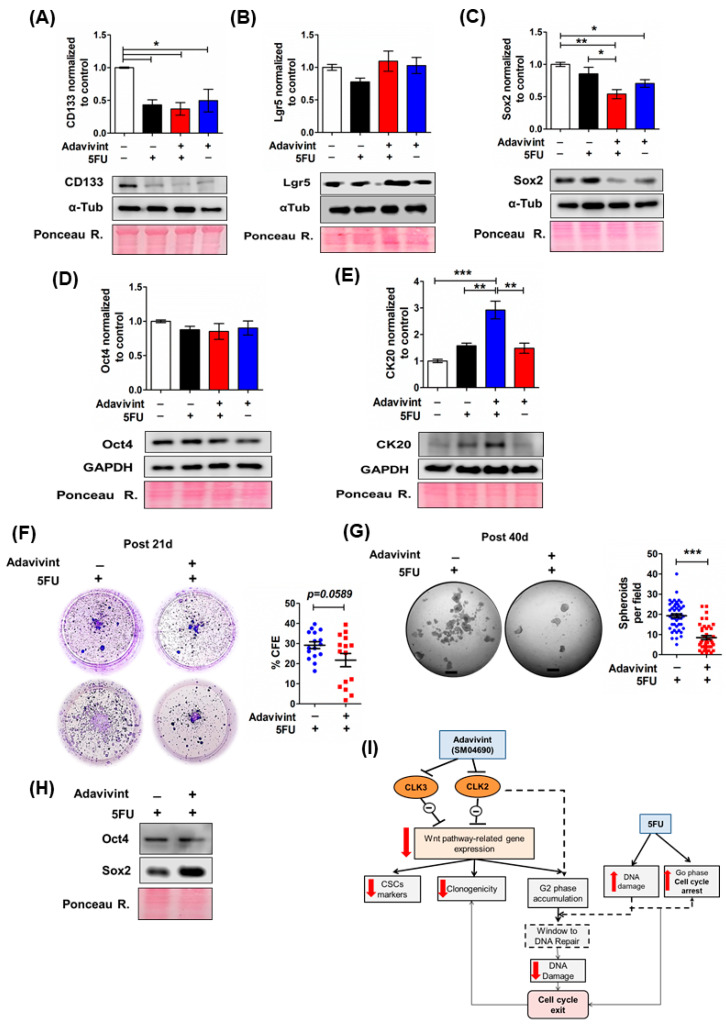
Adavivint Wnt inhibitor reduces clonogenicity and cancer stem-marker expression in SW620 spheroids. (**A**–**D**) Representative immunoblots of cancer stem cell markers CD133 (**A**), Lgr5 (**B**), Sox2 (**C**), and Oct4 (**D**) of SW620 spheroids treated with or without Adavivint for 72 h and subsequently with or without 5FU for 72 h. (**E**) Representative immunoblot of cytokeratin 20 (CK20) of SW620 spheroids treated as described in (**A**). Ponceau Red staining for total protein was used as loading control. Quantification of the relative levels of indicated proteins was normalized to control-treated with vehicles of Adavivint and 5FU. Data are represented by the mean ± SEM of at least three independent tests. The ability to regrowth of SW620 cells from spheroids treated with 5FU alone or in combination with Adavivint was evaluated in anchorage-dependent (**F**) and independent conditions (**G**). (**F**) The clonogenic capacity was evaluated on the 21st-day post-treatment in dependent-anchorage conditions (% CFE, colony formation efficiency). (**G**) The spheroid-forming capacity was determined by counting spheroids in each field of photographed wells on the 40th day of post-treatment (free-drug time) (Scale bar: 500 μm) (n: 3). (**H**) We collected the spheroids established on the 40th day of post-treatment, and levels of Oct4 and Sox2 were determined (n: 1). Ponceau Red staining for total protein was used as loading control * *p* < 0.05; ** *p* < 0.001; *** *p* < 0.0001. (**I**) Model of Adavivint’s mechanism as a Wnt pathway inhibitor in SW620 spheroids and its effect on 5FU-induced DNA damage. Adavivint inhibits the canonical Wnt pathway by CDC-like kinase (CLK) activity inhibition (CLK2 and CLK3), leading to the formation of unstable transcripts of Wnt pathway-related genes. Canonical Wnt signaling inhibition decreases the expression of CSCs markers (such as Sox2 and CD133) and clonogenic capacity. Adavivint induces G2 phase accumulation, in which cells can repair damaged DNA (endogenous and genotoxic damage caused by 5FU). The balance between Cyclin-CDK complexes and CDK inhibitors at the end of the mitotic phase could be decisive in re-entering the cell cycle after DNA repair. Adavivint reduced c-Myc levels, which suggests a disturbed expression of proteins involved in cell cycle progression. We can rule out the effect of CLK inhibition on other cell signaling pathways. The dotted line depicts the suggested mechanism based on published data.

## Data Availability

The data are available with this article.

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
