# Peer review of "Canonical Wnt Pathway Is Involved in Chemoresistance and Cell Cycle Arrest Induction in Colon Cancer Cell Line Spheroids"

_ijms, 2023, doi:10.3390/ijms24065252_

Round 1
Reviewer 1 Report
In their manuscript Moreno-Londoño et al. investigated the role of the canonical Wnt/β-catenin pathway in CSCs resistance to 5-FU treatment in colon tumor spheroids in in vitro model. The authors revealed that 5-FU induces in all CRC spheroids tested cell death, DNA damage, and quiescence. Additionally, they found that the activation of canonical Wnt signaling promotes survival in different spheroid cell contexts, which counteracts the negative effects induced by 5-FU.
The experimental design has been properly planned and results have been well documented.
I have only minor observations:
1. Lines 81-82. Please complete sentence: “Our data suggest, therefore, that canonical Wnt signaling is involved in ..”
2. The human RKO, SW480 and SW620 cell lines were used, were purchased or gifted? This information should be added to “4.3. Cell lines” section.
3. Quality of the manuscript will further increase if authors, on the basis of results obtained, include molecular mechanism of 5-FU action on Wnt pathway.
Author Response
In their manuscript Moreno-Londoño et al. investigated the role of the canonical Wnt/β-catenin pathway in CSCs resistance to 5-FU treatment in colon tumor spheroids in in vitro model. The authors revealed that 5-FU induces in all CRC spheroids tested cell death, DNA damage, and quiescence. Additionally, they found that the activation of canonical Wnt signaling promotes survival in different spheroid cell contexts, which counteracts the negative effects induced by 5-FU.
The experimental design has been properly planned and results have been well documented.
I have only minor observations:
- Lines 81-82. Please complete sentence: “Our data suggest, therefore, that canonical Wnt signaling is involved in ..”
The sentence was completed.
- The human RKO, SW480 and SW620 cell lines were used, were purchased or gifted? This information should be added to “4.3. Cell lines” section.
All cells were obtained from American Type Culture Collection (ATCC) (Manassas, VA, USA) and were authenticated by Short Tandem Repeat DNA profiling analysis performed at the Instituto Nacional de Medicina Genómica (INMEGEN) in Mexico City. We have added this information in the “4.3 Cell lines” section.
- Quality of the manuscript will further increase if authors, on the basis of results obtained, include molecular mechanism of 5-FU action on Wnt pathway.
Following your suggestion, we have now included a graphical abstract of our results in Figure 7.
Reviewer 2 Report
This manuscript is well written, but the experiments purporting to link wnt signaling to the action of 5-FU on cancer stem cells and cancer responses are sometimes confusing and often not definitive. Both the title and the first part of the abstract refer to cancer spheroids and cancer stem cells; in my opinion this implies that primary or even metastatic cancer stem cells are studied, however, this study focuses on two human cancer cell lines: SW480, RKO and SW620 and some transfected variants. This is fine, but it should be made clear from the title and the start of the abstract. The evidence that these cell lines propagate via a minority stem cell population is not strong. The title would be more informative if it concluded …. colon cancer cell line spheroids. The Abstract should be rewritten to remove cancer stem cells and replace the reference to cancer stem cells with SW480, RKO or SW620. Citation of previous work showing 5FU kills SW480, RKO and SW620 cells should be prominent in the introduction.
There are details that need to be addressed:
1. The abstract states: “Activating the canonical Wnt pathway with Wnt3a in 24 RKO spheroids decreased the 5-FU-induced cell death.” I couldn’t see the survival curves for RKO +/- Wnt3a.
In figure 4. The results for the markers of survival or apoptosis are presented, but the results are not definitive.
2. In figure 1 A there is something seriously wrong with the statistics for the SW480 spheroids – the %s add up to more than 100 %. Also there seems to have been more cells counted than for the 2D cultures – I am not sure that this gives an appropriate representation of the results. CD44 and CD133 expression are known to vary with culture conditions, they are not definitive markers for stem cell populations. It is not clear that LGR5 expression is a marker for stemness in cell lines such as SW480, RKO or SW620.
3. Adavivant inhibits at 20nM, yet the effects on growth and b-cat in these cell lines occur at 1μM, at lower concentrations e.g. 100nM the drug stimulates b-cat. Some indication that this might be due to off target effects of the drug should be given. The removal of b-cat with the DNTCF4 contract only leads to a little more killing with 5-FU (Fig 5B)
4. The final sentence of the abstract suggests the combined treatment might enhance resistance to 5FU. The title to Fig 6 is “Canonical Wnt pathway inhibition in CRC spheroid increases 5FU resistance by decreasing cell death and inducing cell cycle arrest”, but no data on cell death is shown and the cell cycle results for the combination are similar to the 5-FU alone. The data in Fig 7 do not definitively support the concept that a 5-FU resistant clone emerges as a result of treatment with the combined inhibitors.
Author Response
This manuscript is well written, but the experiments purporting to link wnt signaling to the action of 5-FU on cancer stem cells and cancer responses are sometimes confusing and often not definitive. Both the title and the first part of the abstract refer to cancer spheroids and cancer stem cells; in my opinion this implies that primary or even metastatic cancer stem cells are studied, however, this study focuses on two human cancer cell lines: SW480, RKO and SW620 and some transfected variants.
This study focuses on three human cancer stem cells obtained from three cancer cell lines. As we described in the Material and Methods section, RKO and SW480 cell lines were isolated from primary cancer, whereas the SW620 cell line was derived from a lymph node metastasis from the same patient that the SW480 cell line was derived. RKO cell line is a prototype of BRAF-driven colon cancer cells that express wild-type p53 and exhibit highly inducible canonical Wnt signaling with no basal activation. SW480 and SW620 cell lines are the prototypes of KRAS-driven cancer cells. They express KRAS and p53 mutated proteins (TP53 R273H, KRAS G12V) and display a constitutive active canonical Wnt signaling because they express a truncated APC protein.
This is fine, but it should be made clear from the title and the start of the abstract. The evidence that these cell lines propagate via a minority stem cell population is not strong.
We did not use the whole population of each cell line, but instead, we focused on studying the cancer stem cell subpopulation using tumorsphere culture. Colon spheroids have been widely described as a stable in vitro model to study stem cells since their enrichment has been demonstrated in these cultures. To establish whether RKO, SW480 and SW620 colon cancer cell lines have the capacity to grow in non-adherent conditions, cultures in a clonal density upon anchorage-independent growth and serum-free medium were performed, and the spheroids formation efficiency (SFE) was quantified. In each cell line, a cellular subpopulation survived and generated spheroids. The self-renewal capacity is recognized through consecutive cultures, where spheroids from the first generation are dissociated at cell single and seeded to form spheroids in a subsequent generation and so on. As can be observed in Figure 1H, the quantification of SFE in the first, second, third, and fourth generations showed that RKO, SW480, and SW620 cells had a cellular subpopulation able to maintain the capacity to form spheroids (Figure 1H). This strongly supported the notion of the self-renewal process and then that the spheroids are enriched in cancer stem cells.
The title would be more informative if it concluded …. colon cancer cell line spheroids. The Abstract should be rewritten to remove cancer stem cells and replace the reference to cancer stem cells with SW480, RKO or SW620.
We consider that both the title and the Abstract are accordingly with the main findings described in our work.
Citation of previous work showing 5FU kills SW480, RKO and SW620 cells should be prominent in the introduction.
Our work is not focused on showing 5FU cancer cell- killing properties but on understanding how colon cancer cells are resistant to 5FU treatment. Thus, the prominent citation in the introduction is about 5FU resistance promotion in colon cancer patients.
There are details that need to be addressed:
- The abstract states: “Activating the canonical Wnt pathway with Wnt3a in 24 RKO spheroids decreased the 5-FU-induced cell death.” I couldn’t see the survival curves for RKO +/- Wnt3a.
You couldn’t see the survival curves because we did not show survival curves in which negative effects could be due to either an increase in apoptosis, or a decrease in cell proliferation. We directly measured the effect of 5FU treatment on cell death by examining the apoptosis levels by detecting the apoptosis marker cleaved PARP (cPARP) and the expression of anti-apoptotic proteins such as survivin and Bcl-2.
In figure 4. The results for the markers of survival or apoptosis are presented, but the results are not definitive.
The results presented are representative of at least three independent experiments using different cell preparations and are statistically significant, indicating that Wnt3a decreased by 20% the cell death induced by 5FU (Figure 4C) and that both Wnt3a and Wnt5a ligands increased almost twice the expression of anti-apoptotic proteins such as survivin and Bcl-2 (Figure 4D) compared to RKO spheroids treated with 5FU alone.
- In figure 1 A there is something seriously wrong with the statistics for the SW480 spheroids – the %s add up to more than 100 %. Also there seems to have been more cells counted than for the 2D cultures – I am not sure that this gives an appropriate representation of the results. CD44 and CD133 expression are known to vary with culture conditions, they are not definitive markers for stem cell populations. It is not clear that LGR5 expression is a marker for stemness in cell lines such as SW480, RKO or SW620.
In Figure 1A, there is no wrong with the statistics for the SW480 spheroids: the % adds up to 100% in 2D and 99.74% in spheroids (3D).
Regarding CD44 and CD133 expression, you are right that they vary with culture conditions. That is why stem cells must be characterized not only based on cell markers expression but also on functional assays as clonogenic ability, as we used here to probe that spheroids are enriched in stem cells since they maintained spheroid forming ability levels for more than 3 generations.
LGR5 expression has been widely reported as a classical stem cell marker of intestinal stem cells.
- Adavivint inhibits at 20nM, yet the effects on growth and b-cat in these cell lines occur at 1μM, at lower concentrations e.g. 100nM the drug stimulates b-cat. Some indication that this might be due to off target effects of the drug should be given.
Following your suggestion, we have explained better at the Results and discussion sections the mechanisms of action of Adavivint and that despite it being a potent inhibitor of canonical Wnt related genes expression via a β-catenin-independent mechanism, mainly through CLK2 and CLK3 inhibition resulting in unstable transcripts of Wnt related genes, we cannot rule out the impact over other signaling pathways that could explain the overall anti-tumor mechanism of this drug. For example, Deshmukh V et al. (2019) demonstrated that Adavivint also disturbs STAT3, NFKB, and FOXO1 signaling, reducing inflammatory processes and improving chondrocyte function during osteoarthritis treatment [32]. In Cancer, CLK2 is involved in proliferation, survival, and poor disease prognosis. CLK2 activity inhibition by Adavivint could impact negatively the proliferation of cells in spheroids regardless of its effect on canonical Wnt signaling.
The removal of b-cat with the DNTCF4 contract only leads to a little more killing with 5-FU (Fig 5B)
Beta-catenin is not removed with the expression of the DNTCF4 and has a constitutive expression in cells. When the DNTCF4 is expressed, the interaction between b-catenin with TCF4 to activate transcription of canonical Wnt target genes is impaired, and thus their transcription is blocked.
- The final sentence of the abstract suggests the combined treatment might enhance resistance to 5FU. The title to Fig 6 is “Canonical Wnt pathway inhibition in CRC spheroid increases 5FU resistance by decreasing cell death and inducing cell cycle arrest”, but no data on cell death is shown and the cell cycle results for the combination are similar to the 5-FU alone.
Data on cell death is shown in Figure 6A. The apoptosis marker cPARP (visualized as cleaved PARP) was clearly diminished in spheroids of both cell lines as a result of canonical Wnt inhibition with Adavivint in combination with 5FU. Regarding the cell cycle arrest, it can be observed in Figure 6E that, as expected, while 5FU increased the proportion of cells in Go and its combination with Adavivint did not modify this, Adavivint alone or in combination with 5FU induced G2 phase accumulation resulting in deep cell cycle arrest after treatment.
The data in Fig 7 do not definitively support the concept that a 5-FU resistant clone emerges as a result of treatment with the combined inhibitors.
The data presented in Figure 7 shows that canonical Wnt inhibition with Adavivint alone or in combination with 5FU produces a severe negative impact on stemness-relevant phenotype: diminished the clonogenic ability and cancer stem cell markers expression in metastatic SW620 spheroids. In these experiments, we seeded cells in low-attachment microplates to evaluate their forming spheroid capacity after treatment with 5FU alone or in combination with Adavivint and followed the cultures for up to 40 days. As can be seen in Figure 7G, although we found a great depletion in the number of spheroids formed, few cells persisted and formed established spheroids after 40 days of culture that remarkably showed an increase in the expression levels of the stem cell marker Sox2, which suggests an increase in proliferation rate, consistent with the emergence of this survival subpopulation with ability to forming spheroids (Figure 7H).
In agreement with our results, very recently Alvarez-Varela A from Edouard Batlle group [42, 43] has revealed that drug-tolerant persister colorectal cancer cells that mediate relapse after chemotherapy downregulate the Wnt/stem cell gene program immediately after chemotherapy. Interestingly, while widespread evidence has demonstrated that CRC growth is driven by a subset of LGR5+ stem-cell-like tumor cells, their analyses revealed that high-resistant cancer cells (HRCs) represent a distinct cell population. Their findings revealed that the adaptation of cancer stem cells to suboptimal niche environments protects them from chemotherapy and identifies a candidate cell of origin of relapse after treatment in CRC that they called HRC. Similar findings have been obtained by Sara M. Weis1 & David A. Cheresh, studying how microenvironmental stresses affect tumor initiation and progression of aggressive pancreatic cancer cells [44]. These authors have recently reported that the harsh microenvironments of the body can push certain pancreatic cancer cells to overcome the stress of being isolated and make them more adept at initiating and forming new tumor colonies. And interestingly, they also observed that while tumor-initiating cells or cancer stem cells (TICs) may inherently possess resistance to such stresses, they showed that a few non-TICs might undergo adaptive reprogramming giving rise to new colonies that will eventually outgrow [44].
Reviewer 3 Report
The authors discuss their study looking at colorectal cancer spheroids and how the Wnt/beta catenin pathway is involved with resistance to 5-FU treatment. They inhibit Wnt signalling using Adavivint alone and in combination with 5-FU. They see differences between the SW480 spheroids and the metastatic derivative (SW620).
Comments:
‘re-growth after 5-FU treatment’ is discussed throughout the manuscript and should be changed to ‘the highest ability for regrowth after 5-FU treatment’ or ‘the highest ability to regrow after 5-FU treatment’ (line 24).
Line 36 – please indicate some statistics for the CRC having the highest incidence and mortality worldwide
Line 81 – the Introduction stops in the middle of a sentence and so some of the aims of the study are missing
Line 90 – should it be CD44v6?
Line 111 – Figure 1H, it mentions that the SFE increased from the second generation but only SW480 cells exhibited a significant increase – SW620 cells show a decrease in SFE in 2nd/3rd/4th generations and this is not clear from the text
Line 163 – higher levels of c-PARP and p-Æ´H2AX are seen in SW480 and SW620 spheroids as well as RKO spheroids and this is not stated in the text
Line 174 – Figure 2D – 5-FU treatment of SW480 spheroids leads to a decrease in the levels of p21 which is statistically significant and this is not clear from the text
Figure 3C – the ‘V’ panel is not present for 21 days?
Line 296 – survivin and not surviving
Figure 4B – please quantify the changes in protein expression for these Western blots; from the blots themselves, it is hard to see that the levels of c-Myc, cyclin D1 and survivin are increasing
Figures 4C/4D and associated text – please clarify further the pretreatment and cotreatment routes as this is not clear
Line 370 – how many passages were needed for restoration of the reporter activity?
Figure 5E – Frequency instead of frecuency on graph y-axes
Lines 559-568 – this summary may be better at the start of the Discussion
Section 4.4 – please refer to a previous paper or add more details on spheroid generation
Line 733 – what % gels were used?
Line 734 – w/v for nonfat dry milk and BSA %s
Author Response
Comments:
‘re-growth after 5-FU treatment’ is discussed throughout the manuscript and should be changed to ‘the highest ability for regrowth after 5-FU treatment’ or ‘the highest ability to regrow after 5-FU treatment’ (line 24).
We have changed to “the highest ability for re-grow after 5-FU treatment”.
Line 36 – please indicate some statistics for the CRC having the highest incidence and mortality worldwide
Done.
Line 81 – the Introduction stops in the middle of a sentence and so some of the aims of the study are missing
The sentence was completed.
Line 90 – should it be CD44v6?
You are right. It should be CD44v6. Thank you.
Line 111 – Figure 1H, it mentions that the SFE increased from the second generation but only SW480 cells exhibited a significant increase – SW620 cells show a decrease in SFE in 2nd/3rd/4th generations and this is not clear from the text
You are right. SW620 cells exhibited a slight reduction in SFE along the passages, but it was not statistically significant. We clarified this in the text.
Line 163 – higher levels of c-PARP and p-Æ´H2AX are seen in SW480 and SW620 spheroids as well as RKO spheroids and this is not stated in the text
We clarified in the results that 5FU induced cell death and DNA damage in all cell lines tested, but RKO-spheroids exhibit higher levels of c-PARP and p-γH2AX as apoptosis and DNA damage markers, respectively, followed by SW480-spheroids and SW620-spheroids.
Line 174 – Figure 2D – 5-FU treatment of SW480 spheroids leads to a decrease in the levels of p21 which is statistically significant and this is not clear from the text
You are right. We clarified in the text that 5FU treatment caused a decrease in p21 levels in SW480-spheroids, whereas no changes were detected in SW620-spheroids.
Figure 3C – the ‘V’ panel is not present for 21 days?
In the clonogenicity assay, we did not show control images corresponding to the 21st day under anchorage-dependent conditions because after the 14th day, the cells reached confluency, and then the culture was not continued for more days.
Line 296 – survivin and not surviving
We replaced surviving by survivin.
Figure 4B – please quantify the changes in protein expression for these Western blots; from the blots themselves, it is hard to see that the levels of c-Myc, cyclin D1 and survivin are increasing
Done. The changes were calculated normalizing with respect to the treatment time (24 h or 48 h), and the values are shown for those blots in Figure 4B.
Figures 4C/4D and associated text – please clarify further the pretreatment and cotreatment routes as this is not clear
The pretreatment and co-treatment protocols were better explained now in figure 4 legend.
Line 370 – how many passages were needed for restoration of the reporter activity?
Cells already had recovered the 90% of reporter activity compared to control cells at the fourth passage.
Figure 5E – Frequency instead of frecuency on graph y-axes
Done
Lines 559-568 – this summary may be better at the start of the Discussion
We made shorter the enumeration of the most relevant findings obtained at the end of the Results section avoiding any discussion of them.
Section 4.4 – please refer to a previous paper or add more details on spheroid generation
We have amplified the information on spheroid generation in the Methods section.
Line 733 – what % gels were used
We added this information in the Materials and Methods section.
Line 734 – w/v for nonfat dry milk and BSA %s
Done. We add this information in Materials and Methods section 4.6
Reviewer 4 Report
In the manuscript entitled “Canonical Wnt pathway is involved in chemoresistance and cell cycle arrest induction in colon cancer spheroids” Moreno-Londoño and colleagues study the role of canonical Wnt signalling in resistance to 5-FU in colon cancer spheroids, using the RKO (sensitive to 5-FU, wild-type APC, BRAF and PIK3CA mutant) and SW480 and its derivative SW620 (resistant to 5-FU, mutant APC –constitutively active Wnt/b-cat–, KRAS and TP53 mutant) cell lines.
The figures are in general clear and support the conclusions drawn by the authors. However, I have some general comments:
1. In protein quantifications, in the figure legends it explains that quantification is “relative” but does not specify to what, is it quantified to loading controls Tubulin or GAPDH? Please specify. Moreover, in Figure 7H, said controls of Tubulin or GAPDH missing.
2. Regarding the original images file:
Figure 4B gels are missing.
Figure 5 gels are missing. Tubulin in Figures 5C and 5D seem to be the same, while the Ponceau staining is not. Please revise them.
Figure 6B and 6D: Tubulin panel doesn’t seem to correspond according to original images (which are the same in both cases in the original image file and not on the manuscript figure, please revise carefully).
Figure 7B: Lgr5 images don’t seem to match any of the original image files. Please revise.
3. Then I have some concerns with some affirmations or maybe their wording.
a. First, I think it would be helpful (although stated in the material and methods), to explain throughout the text that the RKO cells are canonical Wnt wild type and therefore can be activated by adding Wnt ligands to the medium, while SWs bear APC mutations, therefore rendering them constitutively active for canonical Wnt/beta-catenin. In these two different contexts, it is expected to have different outcomes to Wnt modulation, even somehow paradoxical, which I think would be nice to explain / discuss more deeply. Which leads to point number 2 (see below).
b. For instance, the affirmation “canonical Wnt signaling induces CSCs resistance to 5-FU treatment” (Line 650) applies to RKO cells, but it is not clear that it applies to SW cells who already possess 5FU resistance, as inhibition of Wnt signalling by either DNTCF4 or Adavivint results in the opposite: it also increases survival of these cells! Therefore, I suggest the authors extend these affirmations and make them more clear, explaining in which context they remain true.
c. The following affirmation “β-catenin/TCF activation dependent and-independent Wnt pathways have a synergist role in cell survival” (Line 606) is not supported by the data shown, as no experiments to study the synergy between both signalling branches have been performed.
d. “other mechanisms participate in resistance induction to 5-FU treatment of CSCs” (Line 568) I recommend the authors take this opportunity to expand what other pathways could be involved, but also to explain the mechanism of action of Adavivint and its other possible targets that could explain the results they obtained. For instance, the effect on p-CDK2 decrease by dnTCF4 seems to be negligible in comparison to what is observed with Adavivint. Could the effects seen with Adavivint not be entirely due to Wnt inhibition, but also directly due to the inhibition of CLK2?
Minor comments:
Overall: 5-FU and 5FU hyphen inconsistencies, as well as in Wnt-3a and Wnt3a, and re-growth regrowth; misuse of the word “growth” or “regrowth” as a verb (Lines 24, 29, 80, 191, 270, 551, 567, 592, 614, 660, 835, 898, ) grow or regrow should be use instead. Please revise carefully.
Line 10: Mésico to be corrected to “México”
Line 81: unfinished sentence
Line 296: surviving to be corrected to “survivin”
Line 385: SM04590 (Adavivint reference) to be corrected to SM04690
Line 493: show to be corrected to “shown”
Line 606: and-independent please revise hyphen placement
Line 649: free-drug to be changed to drug-free
Line 656: “amazingly” – I recommend the authors do not use this expression
Author Response
In the manuscript entitled “Canonical Wnt pathway is involved in chemoresistance and cell cycle arrest induction in colon cancer spheroids” Moreno-Londoño and colleagues study the role of canonical Wnt signalling in resistance to 5-FU in colon cancer spheroids, using the RKO (sensitive to 5-FU, wild-type APC, BRAF and PIK3CA mutant) and SW480 and its derivative SW620 (resistant to 5-FU, mutant APC –constitutively active Wnt/b-cat–, KRAS and TP53 mutant) cell lines.
The figures are in general clear and support the conclusions drawn by the authors. However, I have some general comments:
- In protein quantifications, in the figure legends it explains that quantification is “relative” but does not specify to what, is it quantified to loading controls Tubulin or GAPDH? Please specify. Moreover, in Figure 7H, said controls of Tubulin or GAPDH missing
We clarified in each figure legend when we used Tubulin, GAPDH, or Ponceau Red staining as the loading control. Regarding Figure 7H, there is no blot for Tubulin or GAPDH. We used Ponceau Red staining for total protein as the loading control.
- Regarding the original images file:
Figure 4B gels are missing
Done. Missing gels were added.
Figure 5 gels are missing. Tubulin in Figures 5C and 5D seem to be the same, while the Ponceau staining is not. Please revise them
Done. Missing blots were added, and we corrected the tubulin of each blot in Figure 5.
Figure 6B and 6D: Tubulin panel doesn’t seem to correspond according to original images (which are the same in both cases in the original image file and not on the manuscript figure, please revise carefully)
Done. We corrected the tubulin of each blot in Figure 6.
Figure 7B: Lgr5 images don’t seem to match any of the original image files. Please revise.
The missing blot was added in the original image file.
- 3. Then I have some concerns with some affirmations or maybe their wording.
- First, I think it would be helpful (although stated in the material and methods), to explain throughout the text that the RKO cells are canonical Wnt wild type and therefore can be activated by adding Wnt ligands to the medium, while SWs bear APC mutations, therefore rendering them constitutively active for canonical Wnt/beta-catenin. In these two different contexts, it is expected to have different outcomes to Wnt modulation, even somehow paradoxical, which I think would be nice to explain / discuss more deeply. Which leads to point number 2 (see below).
RKO cell line is a prototype of BRAF-driven colon cancer cells that exhibit highly inducible canonical Wnt signaling with no basal activation. SW480 and SW620 cell lines are the prototypes of KRAS-driven cancer cells. They express KRAS and p53 mutated proteins (TP53 R273H, KRAS G12V) and display a constitutive active canonical Wnt signaling because they express a truncated APC protein.
- For instance, the affirmation “canonical Wnt signaling induces CSCs resistance to 5-FU treatment” (Line 650) applies to RKO cells, but it is not clear that it applies to SW cells who already possess 5FU resistance, as inhibition of Wnt signalling by either DNTCF4 or Adavivint results in the opposite: it also increases survival of these cells! Therefore, I suggest the authors extend these affirmations and make them more clear, explaining in which context they remain true.
Our data clearly showed that canonical Wnt pathway activation, either induced by ligand (in RKO cells) or already constitutive active (in SW480 and SW620 cells), produced the same effect in colon cancer stem cells: counteracts the negative effects produced in them by 5-FU. This explains why SW480 and SW620 cells are already resistant to 5-FU, and RKO cells are sensitive unless Wnt3a stimulates canonical Wnt in them. In view of these results showing that canonical Wnt activation induces resistance to 5FU-induced negative effects, it would be expected that the inhibition of canonical Wnt would sensitize cells to 5FU treatment. However and unexpectedly, we observed that the inhibition of the b-catenin transcriptional activity, either mediated by the expression of a dnTCF4, or using the inhibitor Adavivint alone or in combination with 5FU also counteracted the negative effect produced by 5-FU alone in cell viability, and enhanced survival in SW480 and SW620 spheroids against 5FU-induced damage. However, although the inhibition of the canonical b-catenin signaling improved the survival of spheroids treated with 5FU by decreasing DNA damage and proliferation, it adversely impacted the stemness phenotype through impaired Sox2 and even Oct4 expression, apparently in dependence on the malignancy grade of cancer cells. Remarkably, we found that combined treatment with 5-FU and Adavivint also induced the survival of a small cell subpopulation, probably through a combined effect on the cell cycle, promoting G0 arrest and G2 phase accumulation. This subpopulation persisted after 40 days of culture that remarkably showed an increase in the expression levels of the stem cell marker Sox2, which suggests an increase in proliferation rate, consistent with the emergence of this survival subpopulation with the ability to form spheroids.
In agreement with our results, very recently Alvarez-Varela A from Edouard Batlle group [42, 43] has revealed that drug-tolerant persister colorectal cancer cells that mediate relapse after chemotherapy downregulate the Wnt/stem cell gene program immediately after chemotherapy. Interestingly, while widespread evidence has demonstrated that CRC growth is driven by a subset of LGR5+ stem-cell-like tumor cells, their analyses revealed that high-resistant cancer cells (HRCs) represent a distinct cell population. Their findings revealed that the adaptation of cancer stem cells to suboptimal niche environments protects them from chemotherapy and identifies a candidate cell of origin of relapse after treatment in CRC that they called HRC. Similar findings have been obtained by Sara M. Weis1 & David A. Cheresh, studying how microenvironmental stresses affect tumor initiation and progression of aggressive pancreatic cancer cells [44]. These authors have recently reported that the harsh microenvironments of the body can push certain pancreatic cancer cells to overcome the stress of being isolated and make them more adept at initiating and forming new tumor colonies. And interestingly, they also observed that while tumor-initiating cells or cancer stem cells (TICs) may inherently possess resistance to such stresses, they showed that a few non-TICs might undergo adaptive reprogramming giving rise to new colonies that will eventually outgrow [44].
- The following affirmation “β-catenin/TCF activation dependent and-independent Wnt pathways have a synergist role in cell survival” (Line 606) is not supported by the data shown, as no experiments to study the synergy between both signalling branches have been performed.
You are right. We have omitted this sentence in line 606 (now in line 628).
- “other mechanisms participate in resistance induction to 5-FU treatment of CSCs” (Line 568) I recommend the authors take this opportunity to expand what other pathways could be involved, but also to explain the mechanism of action of Adavivint and its other possible targets that could explain the results they obtained. For instance, the effect on p-CDK2 decrease by dnTCF4 seems to be negligible in comparison to what is observed with Adavivint. Could the effects seen with Adavivint not be entirely due to Wnt inhibition, but also directly due to the inhibition of CLK2?
Regarding the mechanism of action of Adavivint and its other possible targets: Adavivint (SM04690) is an ATP-competitive small molecule that reduces the expression of TCF1, TCF4 (TCF7L2), AXIN2, and LEF1 at a post-transcriptional level through inhibition of intranuclear kinases CLK2 and DYRK1A thus leading to an overall inhibitory effect on canonical Wnt related genes expression via a β-catenin-independent mechanism. In CRC SM08502 the bioequivalent compound of Adavivint with enhanced oral bio-disponibility, inhibit the Wnt canonical signaling in CRC cell lines mostly through CLK2 inhibition resulting in unstable transcripts of Wnt-related genes (intron retention and exon skipping). Both Adavivint and SM08502 additionally inhibit CLK1 and CLK3 as well as CLK2 in contrast to previously described inhibitors [32, 45]. However, because Adavivint and SM0852 can inhibit several other kinases and CLK is actively involved in pre-mRNA processing, we cannot rule out the impact over other signaling pathways that could explain the overall anti-tumor mechanism of these drugs. We have explained this in the discussion section of the revised version.
Minor comments:
Overall: 5-FU and 5FU hyphen inconsistencies, as well as in Wnt-3a and Wnt3a, and re-growth regrowth; misuse of the word “growth” or “regrowth” as a verb (Lines 24, 29, 80, 191, 270, 551, 567, 592, 614, 660, 835, 898, ) grow or regrow should be use instead. Please revise carefully.
All these inconsistencies were carefully revised and corrected.
Line 10: Mésico to be corrected to “México”
Done
Line 81: unfinished sentence
The sentence was completed
Line 296: surviving to be corrected to “survivin”
Done
Line 385: SM04590 (Adavivint reference) to be corrected to SM04690
It was corrected
Line 493: show to be corrected to “shown”
It was corrected
Line 606: and-independent please revise hyphen placement
We modified the text in this paragraph
Line 649: free-drug to be changed to drug-free
Done
Line 656: “amazingly” – I recommend the authors do not use this expression
Following your recommendation, we changed this expression for “remarkably”.
Round 2
Reviewer 2 Report
The authors have supplied thoughtful responses to most of my comments. I believe the Ms is suitable for publication, but I do not accept the authors equating 2D spheroidal growth with cancer stem cells. I think it is misleading. At the least the title should say ....colon cancer cell line spheroids. Furthermore, I believe the Abstract should summarize the results and avoid the use of cancer stem cells. These cell lines are so far removed from the original primary and metastatic tumors, who knows what their stemness is likely to be today. I still believe that as presented, the Abstract and Title would mislead readers who are looking to improve their understanding of colon cancer stem cell responses to different anti-cancer treatments. Otherwise , I believe the data is clear and discussed appropriately.
Author Response
The authors have supplied thoughtful responses to most of my comments. I believe the Ms is suitable for publication, but I do not accept the authors equating 2D spheroidal growth with cancer stem cells. I think it is misleading. At the least the title should say ....colon cancer cell line spheroids. Furthermore, I believe the Abstract should summarize the results and avoid the use of cancer stem cells……..
Following your suggestion the title was changed to say ….colon cancer cell line spheroids”. The new title is then the following: “Canonical Wnt pathway is involved in chemoresistance and cell cycle arrest induction in colon cancer cell line spheroids”
In the Abstract, it was clearly established that our work aimed to investigate the role played by the canonical Wnt/β-catenin pathway in cancer stem cells resistance to 5FU treatment using tumor spheroids as a model of CSCs enrichment of CRC cell lines. And when the results are summarized, we always use the term “spheroids” avoiding using the term CSCs.
We thank the reviewer for their valuable comments and suggestions to improve our work.
Reviewer 4 Report
Authors have addressed most raised issues and adapted the manuscript accordingly. The quality of the manuscript has improved by adding the missing figures, clarifying the controls used, and extending the discussion and interpretation of the results. I recommend accepting the paper in its current form, after minor spelling mistakes have been corrected from the added text, specifically on page 18 of 28:
Line 674: Eduard instead of Edouard (and maybe mention both first authors of both cited articles, as they differ)
Line 683: Sara M. Weis without the “1”
Line 691: differences instead of differendes
Please revise the added text carefully.
Author Response
Authors have addressed most raised issues and adapted the manuscript accordingly. The quality of the manuscript has improved by adding the missing figures, clarifying the controls used, and extending the discussion and interpretation of the results. I recommend accepting the paper in its current form, after minor spelling mistakes have been corrected from the added text, specifically on page 18 of 28:
Line 674: Eduard instead of Edouard (and maybe mention both first authors of both cited articles, as they differ)
The name was corrected and both first authors were cited.
Line 683: Sara M. Weis without the “1”
Done
Line 691: differences instead of differendes
Done
Please revise the added text carefully.
The added text was carefully revised.
We thank the reviewer for their valuable comments and suggestions to improve our work.